# Comparison of Methods for Measuring Protein Concentration in Venom Samples

**DOI:** 10.3390/ani10030448

**Published:** 2020-03-08

**Authors:** Aleksandra Bocian, Sonja Sławek, Marcin Jaromin, Konrad K. Hus, Justyna Buczkowicz, Dawid Łysiak, Vladimir Petrílla, Monika Petrillova, Jaroslav Legáth

**Affiliations:** 1Faculty of Chemistry, Rzeszow University of Technology, 35-959 Rzeszów, Poland; 153712@stud.prz.edu.pl (S.S.); mjaromin@prz.edu.pl (M.J.); knr.hus@gmail.com (K.K.H.); czaporj@prz.edu.pl (J.B.); 155765@stud.prz.edu.pl (D.Ł.);; 2Department of Anatomy, Histology and Physiology, University of Veterinary Medicine and Pharmacy, Komenského 73, 041 81 Kosice, Slovakia; petrillav@gmail.com; 3Zoological Department, Zoological Garden Košice, Široká 31, 040 06 Košice-Kavečany, Slovakia; 4Department of General Education Subjects, University of Veterinary Medicine and Pharmacy, Komenského 73, 041 81 Kosice, Slovakia; monika.petrillova@uvlf.sk; 5Department of Pharmacology and Toxicology, University of Veterinary Medicine and Pharmacy, Komenského 73, 041 81 Kosice, Slovakia

**Keywords:** protein concentration, snake venom, *Agkistrodon contortrix*, *Naja ashei*

## Abstract

**Simple Summary:**

Snake venom is mostly composed of proteins and peptides, which are of interest to many researchers due to their potential pharmacological properties. Due to their biochemical character, these components are analyzed using proteomic techniques such as electrophoresis, chromatography and mass spectrometry. A very important stage of such studies is the measurement of protein concentration in the sample, which is most often performed by colorimetric methods. In the presented article, we used five such techniques on venoms of two snake species, namely *Agkistrodon contortrix* and *Naja ashei*. In the case of *A. contortrix* venom, four methods provide similar concentration values, whereas, in the case of *N. ashei,* the differences between results are very significant. The source of these differences should probably be seen in the differences in amino acid composition of proteins of these two venoms. With this report, we would like to draw attention to the need to select an appropriate method for measuring the concentration of protein in the venom, especially in the case of Elapid species.

**Abstract:**

Snake venom is an extremely interesting natural mixture of proteins and peptides, characterized by both high diversity and high pharmacological potential. Much attention has been paid to the study of venom composition of different species and also detailed analysis of the properties of individual components. Since proteins and peptides are the active ingredients in venom, rapidly developing proteomic techniques are used to analyze them. During such analyses, one of the routine operations is to measure the protein concentration in the sample. The aim of this study was to compare five methods used to measure protein content in venoms of two snake species: the Viperids representative, *Agkistrodon contortrix,* and the Elapids representative, *Naja ashei*. The study showed that for *A. contortrix* venom, the concentration of venom protein measured by four methods is very similar and only the NanoDrop method clearly stands out from the rest. However, in the case of *N. ashei* venom, each technique yields significantly different results. We hope that this report will help to draw attention to the problem of measuring protein concentration, especially in such a complex mixture as animal venoms.

## 1. Introduction

Snake venom evolved to immobilize and capture the prey as well as for defense against other predators. The composition of snake venom depends mainly on the species, but also age, sex, and season [1,2]. The main components of venom include proteins, with and without enzymatic activity, and peptides. In addition, venom contains marginal amounts of nucleosides, sugars, lipids, and inorganic ions, which are less interesting from a pharmacological point of view, as it is the protein-peptide origin that is responsible for venom toxicity [3]. Venom proteins have anticancer [4], analgesic [5], antibacterial [6] and cardiovascular [7] properties and are therefore considered to have great pharmacological potential. As such, venom components are of wide interest to many research groups and have been the subject of extensive research. As about 90% of venom components are of peptide origin, all venom analyses use increasingly fast developing proteomic techniques, including electrophoresis [8,9,10,11], immunodetection techniques [12,13,14,15], different types of chromatography [16,17,18,19] and mass spectrometry [20,21,22,23]. Each proteomic method has specific requirements for both the sample components and the protein concentration range. This is particularly important for 2D electrophoresis, where isoelectric focusing strips have a specific protein "capacity" and chromatographic techniques, where column capacity is important. In general, two approaches are used in the analysis of venom proteomes. In the first one, the venom is freeze-dried or dried and suspended in a specific amount of buffer to obtain samples at a specific concentration [24,25,26,27]. The second approach uses crude venom and this approach requires a step to measure the protein concentration of the sample [28,29,30]. 

Colorimetric methods are most commonly used in estimating the protein concentration of the sample; however, there are also methods based on protein fluorescence or the direct measurement of sample absorbance at 280 nm without any protein reference [31]. Moreover, recently, a new technique for absolute protein quantitation hasalso emerged which utilizes ICP-MS [32]. The colorimetric techniques include the Bradford assay, the copper ion reduction method or the Biuret method with its modifications e.g., with bicinchoninic acid (BCA), while the fluorimetric techniques proposed by ThermoFisher include the Qubit^®^ (Thermo Fisher Scientific, Waltham, MA, USA) method. Each of these methods is based on a different mechanism of action and takes into account other aspects characterizing the components of the sample. Therefore, we evaluated how the different protein composition of venom samples affects the measurement results. Two species were selected for analysis: *Agkistrodon contortrix*, a representative of Viperids, and *Naja ashei*, a representative of Elapids. The venoms of Elapids and Viperids species differ strongly in terms of protein composition. Elapids are characterized by a decidedly predominant amount of 3FTx proteins and phospholipases A_2_ (PLA_2_s) and low content of other components such as metalloproteinases (SVMPs), CRISP proteins, L-amino acid oxidases (LAAOs) or in some cases Kunitz type protease inhibitors. On the other hand, the venoms of Viperid species also have a high PLA_2_s content, in comparison, i.e., a relatively low content of CRISPs and LAAOs, but the predominant groups are usually SVMPs and serine proteases (SVSPs), while 3FTx proteins are not observed [33]. The aim of this experiment was to compare five techniques for measuring protein concentration: the Bradford assay, BCA assay, 2-D Quant Kit, Qubit^®^, and direct technique using NanoDrop™ (Thermo Fisher Scientific, Waltham, MA, USA) in the venoms of two snake species described earlier for their protein composition.

## 2. Materials and Methods 

Both venoms were extracted in the breeding garden Pata near Hlohovec (Slovakia, veterinary certificate No. CHEZ-TT-01), withthe permission of the State Nature Protection of the Slovak Republic No. 03418/06 and No. 237/2002 Z.z. The same samples were used in our earlier proteomic research [34,35]. 

Three ready-to-use kits for measuring protein concentration in solutions were used in the study: the Pierce™ BCA Protein Assay Kit (Thermo Fisher Scientific, Waltham, MA, USA; 23225), 2-D Quant Kit (GE Healthcare; Little Chalfont, UK 80-6483-56) and Qubit^®^ Protein Assay Kits (Thermo Fisher Scientific, Waltham, MA, USA; Q33211). Bradford Reagent (BioShop; Burlington, ON, Canada, BRA222) was also used for analysis.

The aqueous dilutions of venom were experimentally selected to fit into the range of the standard curve. 

### 2.1. Pierce™ BCA Protein Assay Kit

The nine-point calibration curve was performed in duplicate from the BSA included in the kit in the concentration range from 0 to 2000 µg/mL. Dilutions of 500× for *Naja ashei* (Na) and 100× for *Agkistrodon contortrix* (Ac) were used to measure protein concentration in venom. The analyses were performed in triplicate. After a 30-minute incubation of the samples with Working Reagent (37 °C), the absorbance of the samples was measured on a spectrophotometer (Evolution™ 201/220 UV-Visible Spectrophotometer, (Thermo Fisher Scientific, Waltham, MA, USA) at 562nm. 

### 2.2. 2-D Quant Kit

A six-point reference curve from 0 to 50 µg was determined using the BSA included in the kit. Twenty-fold and 10-fold dilutions of Na and Ac, respectively, were used to measure protein concentration in venom. The standard curve was prepared in duplicate and the samples in triplicate. The proteins in both the tested samples and the standard curve was precipitated with the reagents from the set, then dissolved in water and copper solution. After adding color working reagent and waiting 15 minutes, the absorbance of samples was measured at 480 nm. 

### 2.3. Bradford Assay

A six-point calibration curve in the range 0.2 to 2 µg/µL was performed using a BSA solution from the 2-D Quant Kit. Both venoms were diluted 200 times for analysis, 1.5 mL of Bradford reagent was added to all samples and, after 20 min, absorbance was measured at 595nm. The standard curve was prepared in duplicate and the samples in triplicate.

### 2.4. Qubit^®^ Protein Assay Kit

The three-point calibration curve was performed according to the manufacturer’s instructions using the included Qubit™ Standard 1–3 reagents. To perform the analysis, 500 times diluted 10 µL venom samples were combined with 190 µL Qubit™ Working Solution and measured on Qubit^®^Fluorometer ((Thermo Fisher Scientific, Waltham, MA, USA). The standard curve was prepared in duplicate and the samples in triplicate.

### 2.5. NanoDrop

Direct concentration measurement was performed on the NanoDrop™ 2000 ((Thermo Fisher Scientific, Waltham, MA, USA) at 280 nm, with an estimated percent extinction coefficient (ε1%) of 10. The analysis was performed in triplicate on 10 times diluted venoms.

### 2.6. Analysis of Results

The calibration curves and calculations were performed in Microsoft Excel. According to the manufacturer’s instructions for the Pierce™ BCA Protein Assay Kit, a polynomial trend line was determined from the standard curve graph. In the case of the Bradford assay and 2-D Quant kit, a linear trend line was used. The results were analyzed statistically by Statistica software v. 13.3. The protein concentration met the assumptions of normality (Shapiro–Wilk test *p* > 0.05) and homogeneity of variance, so to assess the differences between the groups, a main effects analysis of variance (ANOVA *p* < 0.05) and Tukey HSD (post-hoc *p* > 0.05) test were used.

## 3. Results

A summary of the obtained concentrations is presented in Table 1. 

The comparison of different methods for protein concentration determination in the case of *Agkistrodon contortrix* venom shows that four methods gave similar concentration values that were in the range of 50 µg/ µL. In turn, the protein concentration calculated using the NanoDrop™ method was different from the rest of the values. Tukey HSD test showed that all methods can be classified into three homogenous subsets (BCA and 2-D QK–a; Bradford and Qubit™–b; NanoDrop™–c), where two subsets (a and b) are not significantly different. For *Naja ashei* venom, however, the results are diametrically different and all methods gave statistically significant different results (Figure 1).

## 4. Discussion

The choice of method of measuring the protein concentration is a crucial step in proteomic analyses but is frequently neglected in studies involving venom. Very often, the selection of methods is mainly dictated by the speed of analysis, the availability of reagents, or just by habit.

Four of the methods we tested are used to measure the concentration of proteins in venom, but we have not found any work in which Qubit^®^ Protein Assay Kit was used for venom analysis. 

The BCA method provides quantitative information on proteins that are able to reduce copper ions from Cu^2+^ to Cu^+^ through the presence of cysteine and tryptophan. [36]. Therefore, the measurement results will be significantly influenced by the primary structure of proteins, i.e., how many cysteine and tryptophan residues they contain. This method has been used to measure the concentration of proteins in venom, e.g., in works on the *Naja atra* cobra [37], snakes of the *Agkistrodon* genus [30] and in the study on PLA_2_ interaction with bacteria and cancer cells [38]. In our study, for *A. contortrix* venom, this method gave comparable results to other methods, while for *N. ashei* it gave the third highest result.

In the 2-D Quant Kit method, copper ions attach to the peptide bonds in the protein, so this method is independent of the amino acid composition of the proteins in the sample. This kit has been repeatedly used to measure the concentration of proteins in venom [28,39] and this is also the most commonly used method in our laboratory [10,34,35,40]. However, obtained results by 2-D Quant Kit and Bradford for *A. contortrix* venom show low variability, whereas for *N. ashei* venom, the differences are large. This observation could explain why we have never managed to obtain well-separated and optimized gels for species of the genus *Naja* [40] and *Dendroaspis*, and we have had no problems with Viperids [10,34]. As the gel staining procedure involves the use of Coomassie Brillant Blue, which is also the dye in the Bradford method, the significant difference in the concentration values obtained from Bradford and 2-D Quant kit assays in the case of *Naja ashei* may be the reason why we observed gels that were not stained enough as if we added too few proteins.

The Bradford assay is a colorimetric method for the quantitative determination of proteins using the ability to bind Coomassie Brillant Blue to the amino acids present in proteins such as histidine, lysine, but mainly arginine [41]. Thus, similarly to the BCA method, the content of these amino acids in proteins will influence the results obtained. The Bradford assay has also been used many times to measure protein concentration in venom [8,29,42,43,44,45]. Interestingly, this is the only method that indicates that the concentration of protein in *A. contortrix* venom is higher than in *N. ashei* and this result seems unlikely from our experience.

The direct method of determining protein concentration on the NanoDrop™ spectrophotometer is based on measurement at 280 nm, where tyrosine, tryptophan, and cysteine (mainly cystine) display absorbance. Therefore, similarly to the BCA method, the content of these amino acids in proteins will influence the final result. Moreover, in the A280 method, it is important to know the extinction coefficient (which is a parameter determining the degree of light absorption by a given substance at a particular wavelength) of the measured protein. Usually, in the case of single proteins, this information is available, or at least it can be approximated from the relative amounts of W, Y, and C in the primary sequence of the protein. However, in complex, unknown mixtures, it is hard to correctly estimate this parameter, and thus the calculated values can be very different from the true ones. Nevertheless, this method is commonly used in snake venom studies [46,47,48].

The last method used was the Qubit method, which allows one to quantify the presence of protein by binding fluorophore with hydrophobic side chains of amino acids present in the studied material. However, as a result of pH changes, other amino acids can also be marked if they are present in hydrophobic protein fragments [49]. Therefore, this method seems to be not only dependent on the protein sequence, but also structure. No examples of the use of this technique in proteomic studies of snake venom have been found in the literature. This may be due to the fact that this kit is designed to measure very low concentrations and requires high dilutions for venom. To fit into the measurement range, we had to dilute the venom 500 times.

From the obtained data, it is not easy to decide which method is the most accurate in determining venom protein concentration, as we do not know its true value. However, with *A. contortrix* venom, four methods provided similar results, which were within 50 µg/ µL. Only the method which is based on direct 280 nm absorbance measurements gave a markedly different value. We think that such a significant difference can be caused by our assumption that the extinction coefficient of both venom protein mixtures is 10, which is a standard procedure when this parameter is unknown; however, it could largely influence the final calculated values, especially if the true value of this factor is very different from the accepted one. On the other hand, in the case of *N. ashei* venom, the spread of data is so large that no conclusions can be drawn. It is not entirely clear why these differences arise; however, different primary sequences of proteins that constitute snake venoms could be the main cause, as the available methods for determining protein concentration are highly dependent on the content of individual amino acids in the sample. Thus, it is always advisable to take into account the protein composition of analyzed venom and accept the calculated values with caution. The venom proteome of *A. contortrix* was described in at least two works and both indicated that the highest percentage of venom is represented by phospholipases A_2_, metalloproteases and serine proteases. L-amino acid oxidases, CRISP proteins and lectin-like proteins are less abundant/less often seen [34,50]. On the other hand, 70% of *N. ashei* venom is composed of 3FTx proteins, less than 30% are PLA_2_ and about 2% SVMP, and the remaining groups—CRISP, VNGF (Venom Nerve Growth Factor), nucleases and CVF (Cobra Venom Factor)—occur in trace amounts [35]. Subsequent studies on fractions obtained from chromatographic separation of this venom have shown that LAAO, SVSP and other proteins are also present, but they represent a very small percentage of venom proteins [51]. Therefore, the main difference between the analyzed venoms is a high content of proteases from SVMP and SVSP groups in *A. contortrix* venom, with the absence of 3FTx proteins, which constitute the dominant group in *N. ashei*. It seems that the amino acid composition of *A. contortrix* venom proteins does not affect the measurement results, regardless of whether the test actually measures the amount of cysteine and tryptophan (BCA test) or arginine, tryptophan and lysine (Bradford assay) or peptide bonds. However, in the case of *N. ashei*, where venom is mainly composed of 3FTx proteins, the specificity of the test becomes relevant. This is probably due to the differences in the composition and proportions of the key amino acids present in the proteins of different venoms that are importantfor the calculationsin the individual tests used. Therefore, it seems that in the comparative inter-genera analysis of venoms that are very different in their protein composition, the choice of method for measuring protein concentration can greatly affect the obtained results. On the other hand, in the case of studies concerning venoms with similar compositions, that decision could be of less importance.

With the presented results, we wanted to draw attention to some limitations associated with the available methods of determining venom protein concentration. We believe that our results will be useful when deciding on the right methodof quantitative analysis of proteins, especially for complex mixtures such as venoms.

## Figures and Tables

**Figure 1 animals-10-00448-f001:**
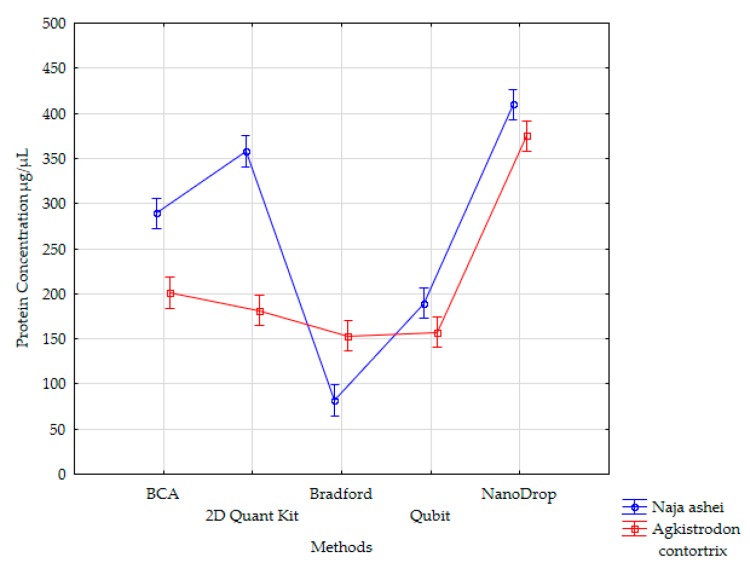
Method means for two-way main effects analysis of variance (Vertical bars denote 0.95 confidence intervals).

**Table 1 animals-10-00448-t001:** Protein concentrations in undiluted venoms of *Naja ashei* and *Agkistrodon contortrix* measured using five different assays.

Venom	Protein Concentration (µg/ µL)			
	Ist Replicate	IInd Replicate	IIIrd Replicate	Mean
	**Pierce™ BCA Protein Assay Kit**			
*Naja ashei*	258.96	291.16	317.68	289.269
*Agkistrodon contortrix*	211.27	190.86	202.01	201.380 ^a^
	**2-D Quant Kit**			
*Naja ashei*	361.38	341.17	370.96	357.837
*Agkistrodon contortrix*	180.69	162.61	201.44	181.578 ^a,b^
	**Bradford Assay**			
*Naja ashei*	81.88	90.73	73.02	81.875
*Agkistrodon contortrix*	151.15	158.96	149.58	153.229 ^b^
	**Qubit^®^ Protein Assay Kit**			
*Naja ashei*	189.50	189.18	190.23	189.637
*Agkistrodon contortrix*	156.83	157.10	157.83	157.254 ^b^
	**NanoDrop™**			
*Naja ashei*	405.05	407.14	417.11	409.77
*Agkistrodon contortrix*	391.52	374.86	358.54	374.97 ^c^

^a,b,c^—each letter indicates homogenous subset of means. The means in each subset are not significantly different.

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
