# Peer review of "Comparison of Methods for Measuring Protein Concentration in Venom Samples"

_animals, 2020, doi:10.3390/ani10030448_

Round 1

Reviewer 1 Report

The mansucript by Bocian et al. demonstrates that the determination of the protein content in venom samples is not trivial. In fact, the results provided by standard methods are highly dependent on the protein structure and therefore can be biased. Manuscript is well written and the interest for thr readership can be high. I recommend publication after dealing with some issues:

1.- Very recently, the use of element mass spectrometry (ICP-MS) has been proposed for the quantification of the absolute amount of proteins in venom samples (Anal. Chem. 2016, 88, 9699−9706). This is a very important application as such absolute quantification can be carried out using generic standards. Therefore it should be mentioned in the introduction becuase this technique could be used as reference in the future in absolute venomics. This is directly linked to the problem highlighted in this article.

2.- As can be seen by the replicate values given in Table 1, precision of the methods ranges from 1 to 10% RSD. Therefore, the number of signiifcant figures associated to the mean values given in Table 1 is excessive

3.- Page 4, line 142: "Much" should be changed by "many"

Author Response

We wanted to thank you very much for your kind words and appreciation of our article. According to the suggestions of both reviewers, we have made the necessary corrections. We have included detailed information about them below. We believe that thanks to the comments and suggestions we managed to improve the quality of our article, for which we thank you very much once again.

The mansucript by Bocian et al. demonstrates that the determination of the protein content in venom samples is not trivial. In fact, the results provided by standard methods are highly dependent on the protein structure and therefore can be biased. Manuscript is well written and the interest for the readership can be high.

I recommend publication after dealing with some issues:

1.- Very recently, the use of element mass spectrometry (ICP-MS) has been proposed for the quantification of the absolute amount of proteins in venom samples (Anal. Chem. 2016, 88, 9699−9706). This is a very important application as such absolute quantification can be carried out using generic standards. Therefore it should be mentioned in the introduction becuase this technique could be used as reference in the future in absolute venomics. This is directly linked to the problem highlighted in this article.

We included above mentioned technique in the introduction.

2.- As can be seen by the replicate values given in Table 1, precision of the methods ranges from 1 to 10% RSD. Therefore, the number of signiifcant figures associated to the mean values given in Table 1 is excessive.

The error has been corrected.

3.- Page 4, line 142: "Much" should be changed by "many"

The error has been corrected.

Reviewer 2 Report

Broad comments by Reviewer 2 to the authors:

An important topic worth publishing on. However, editing is required throughout the ms, particularly to improve the scientific writing style. See detailed comments which I’ve made to help the authors along in this regard. Reference choices could also be improved in some cases (see notes).

One major suggestion I have is to include one more method for determining protein concentration: Nanodrop spectrophotometer. This small machine allows researchers to rapidly (no 30min incubation) determine protein concentrations; thus it is standard practice in venom research (e.g. snake and insect venoms) to use the nanodrop to determine protein concentrations, so this paper would be much more widely useful for the venom field of study if it also compared this method for determining protein concentration. In contrast, the Qubit method was used, despite the authors admitting that “No examples of the use of this technique in proteomic studies of snake venom have been found in the literature” (L170).

Table 1 seems to show how tight the triplicates are for each method on each of the two venoms, but not how accurate the protein measurements are to the actual (ie. true) protein concentration of the two venoms. A helpful addition to the paper would be to order in a known protein concentration (ie. a positive control) calibrator liquid from ThermoFisher, which is used to calibrate NanoDrop machines, and compare the protein concentrations measured across the various methods of that known solution. Note: the calibration solution is only good for about 1hr, as evaporation increases the protein concentration.

How the calibration curves of the various methods relate to Fig.1 is unclear. That is, how do the calibration curves get converted to the one protein concentration shown for each method in Fig.1?

Detailed edits from Reviewer 2 to improve the ms:

Suggested edits to improve the ms:

  • Abstract
    1. L39: reword ‘gives clearly different results’ to saying something more substantial, or at least refer to statistical significance.
    2. L39: add conclusion sentence describing the importance of this work to the venom research field.
  • Intro
    1. Venom also used in defence (e.g. Naja species spitting venom in predator’s eyes; bee venom; etc.)
    2. This ref appears not the best choice for the sentence in which it is referenced: ‘Koh, D.C.; Armugam, A.; Jeyaseelan, K. Snake venom components and their applications in biomedicine. Cell Mol Life Sci 2006, 63, 3030-3041.’
      1. Alternative options: see bottom right of p219 of ‘Casewell, N.R., Wüster, W., Vonk, F.J., Harrison, R.A., and Fry, B.G. (2013). Complex cocktails: The evolutionary novelty of venoms. Trends Ecol. Evol. 28, 219–229.’
    3. L45: add comma after ‘activity’ to avoid confusion.
    4. L45: ‘Besides’ at the start of a sentence is colloquial. Try ‘In addition’
    5. L47: The other components in the venom gland are important for other reasons (e.g. to reduce enzymatic degradation of toxins), so better to reword sentence.
    6. L49: This sentence is not in scientific writing style, but rather colloquial style: ‘No wonder that snake venom continues to be popular among scientists.’
    7. L65-67: non-scientific writing style. (e.g. ‘actually’ and ‘check’)
  • Materials and Methods
    1. L80: “with the permission of”
    2. L93: “in triplicate” instead of “in three repetitions”
    3. L93-95: why the 30min incubation period prior? NanoDrop does not require this. What’s the difference in mechanism of measurement b/w Evolution and Nanodrop? the nanodrop changes the shape of a 2µl droplet on the pedestal and measures the spectra at two different bubble shapes and averages these two and reports this as the protein concentration. Does the Evolution use the same or different method?
    4. L95: Why 562nm? The optical absorbance of proteins is measured most strongly at 280nm. At this wavelength, the absorption of proteins is mainly due to the amino acids tryptophan, tyrosine and cysteine. The varying concentrations of these in different venoms would affect the wavelength of maximum absorption, but I am not sure why 562nm was chosen and that wavelength
    5. L119: state in Methods what level of significance you’re using (ie. p-values <0.05 were considered significant)
  • Results
    1. L127: by ‘statistically similar’, you mean ‘not significantly different’?
    2. L122: change your commas to be full-stops (ie. periods). That is, I suspect that for N. ashei 1st replicate in the Pierce kit, the venom out of the gland (ie. undiluted as the table says) is 258.96 µg/µl (ie. the 96 are the two decimal places after the whole numbers), not ‘258,96’).
    3. 1: units needed for the y-axis protein concentration (ie. µg/µL)
    4. 1: info at the top of the figure is confusing. (e.g. ‘Effective hypothesis decomposition’?). This helpful info from the top of the fig (ie. not all of it; definitely the error bars part) belongs in the figure caption.
  • Discussion
    1. L136: ‘commonly’? In what line of research? Not in venom research in my experience or to my knowledge.
    2. L139: reword ‘we find out about’
    3. L174-176: improve to be scientific writing style
    4. L192: explain why.
    5. L193-195: improve to be scientific writing style

Author Response

We wanted to thank you very much for your kind words and appreciation of our article. According to the suggestions of both reviewers, we have made the necessary corrections. We have included detailed information about them below. We believe that thanks to the comments and suggestions we managed to improve the quality of our article, for which we thank you very much once again.

An important topic worth publishing on. However, editing is required throughout the ms, particularly to improve the scientific writing style. See detailed comments which I’ve made to help the authors along in this regard. Reference choices could also be improved in some cases (see notes).  

One major suggestion I have is to include one more method for determining protein concentration: Nanodrop spectrophotometer. This small machine allows researchers to rapidly (no 30min incubation) determine protein concentrations; thus it is standard practice in venom research (e.g. snake and insect venoms) to use the nanodrop to determine protein concentrations, so this paper would be much more widely useful for the venom field of study if it also compared this method for determining protein concentration. In contrast, the Qubit method was used, despite the authors admitting that “ No examples of the use of this technique in proteomic studies of snake venom have been found in the literature” (L170).

Thank you very much for that suggestion. This method was not originally used by us because we do not have a NanoDrop spectrophotometer. However, we have approached colleagues at another university who provided us with the equipment, so we added new data to the article.We hope that additional results will increase the value of the article.

 Table 1 seems to show how tight the triplicates are for each method on each of the two venoms, but not how accurate the protein measurements are to the actual (ie. true) protein concentration of the two venoms. A helpful addition to the paper would be to order in a known protein concentration (ie. a positive control) calibrator liquid from ThermoFisher, which is used to calibrate NanoDrop machines, and compare the protein concentrations measured across the various methods of that known solution. Note: the calibration solution is only good for about 1hr, as evaporation increases the protein concentration.

As we mentioned earlier, we had to use NanoDrop at another university. Unfortunately our colleagues were not able to provide us with this reagent. Therefore, unfortunately, it is not possible for us to perform the proposed experiment at this point.

How the calibration curves of the various methods relate to Fig.1 is unclear. That is, how do the calibration curves get converted to the one protein concentration shown for each method in Fig.1?

For each method, except NanoDrop, a separate calibration curve was performed according to a given procedure. Protein concentration for a given method was calculated from the corresponding regression function equation based on the measured absorbance of the samples. Calculated values for each method were included in Figure 1.

Detailed edits from CNZdenek to improve the ms:

 Suggested edits to improve the ms:

 1)  Abstract

  1. L39: reword ‘gives clearly different results’ to saying something more substantial, or at least refer to statistical significance.b. L39: add conclusion sentence describing the importance of this work to the venom research field.

The sentence has been rewritten and appropriate summary sentence has been added.

2)  Intro

  1. L43. Venom also used in defence (e.g. Naja species spitting venom in predator’s eyes; bee venom; etc.)

Information has been added.

  1. L44. This ref appears not the best choice for the sentence in which it is referenced: ‘Koh, D.C.; Armugam, A.; Jeyaseelan, K. Snake venom components and their applications in biomedicine. Cell Mol Life Sci 2006, 63, 3030-3041.’
  2. Alternative options: see bottom right of p219 of ‘Casewell, N.R., Wüster, W., Vonk, F.J., Harrison, R.A., and Fry, B.G. (2013). Complex cocktails: The evolutionary novelty of venoms. Trends Ecol. Evol. 28, 219–229.’

Thank you for the suggestion. The reference has been changed.

  1. L45: add comma after ‘activity’ to avoid confusion.

The error has been corrected.

  1. L45: ‘Besides’ at the start of a sentence is colloquial. Try ‘In addition’

The error has been corrected.

  1. L47: The other components in the venom gland are important for other reasons (e.g. to reduce enzymatic degradation of toxins), so better to reword sentence.

The sentence has been rewritten.

  1. L49: This sentence is not in scientific writing style, but rather colloquial style: ‘No wonder that snake venom continues to be popular among scientists.’

The sentence has been rewritten.

  1. L65-67: non-scientific writing style. (e.g. ‘actually’ and ‘check’)

The sentence has been rewritten.

3)  Materials and Methods

  1. L80: “with the permission of”

The error has been corrected.

  1. L93: “in triplicate” instead of “in three repetitions”

The error has been corrected.

  1. L93-95: why the 30min incubation period prior? NanoDrop does not require this. What’s the difference in mechanism of measurement b/w Evolution and Nanodrop? Ie.thenanodrop changes the shape of a 2µl droplet on the pedestal and measures the spectra at two different bubble shapes and averages these two and reports this as the protein concentration. Does the Evolution use the same or different method?

Evolution is a regular spectrophotometer that requires cuvettes for measurements. The 30 minutes of incubation are determined by the procedure included in the Pierce™ BCA Protein Assay Kit.

  1. L95: Why 562nm? The optical absorbance of proteins is measured most strongly at 280nm. At this wavelength, the absorption of proteins is mainly due to the amino acids tryptophan, tyrosine and cysteine. The varying concentrations of these in different venoms would affect the wavelength of

maximum absorption, but I am not sure why 562nm was chosen and that wavelength only.

The procedure was performed in accordance with the manufacturer's instructions included in the Pierce™ BCA Protein Assay Kit (https://www.thermofisher.com/order/catalog/product/23225#/23225).

  1. L119: state in Methods what level of significance you’re using (ie. p-values <0.05 were considered significant)

Information has been added.

4)  Results

  1. L127: by ‘statistically similar’, you mean ‘not significantly different’?

No we can use term ‘not significantly different’. Tukey HSD test showed that four methods can be classified into 2 homogenous subsets (BCA and 2-D QK – a ; 2-D QK, Bradford and Qubit – b ; (both subsets are not significantly different p > 0.05);, because method 2-D QK belongs to both subsets we use term statistically similar .

  1. L122: change your commas to be full-stops (ie. periods). That is, I suspect that for N. ashei 1 st replicate in the Pierce kit, the venom out of the gland (ie. undiluted as the table says) is 258.96 µg/µl (ie. the 96 are the two decimal places after the whole numbers), not ‘258,96’).

The error has been corrected

  1. Fig.1: units needed for the y-axis protein concentration (ie. µg/µL)

The units has been added.

  1. Fig. 1: info at the top of the figure is confusing. (e.g. ‘Effective hypothesis decomposition’?). This helpful info from the top of the fig (ie. not all of it; definitely the error bars part) belongs in the figure caption.

The info has been deleted. Information about error bars has been added in figure description.

5)  Discussion

  1. L136: ‘commonly’? In what line of research? Not in venom research in my experience or to my knowledge.

The sentence has been rewritten.

  1. L139: reword ‘we find out about’

The sentence has been rewritten.

  1. L174-176: improve to be scientific writing style

The sentence has been rewritten and other sentences explaining additional results have been added.

  1. L192: explain why.

An explanation has been added.

  1. L193-195: improve to be scientific writing style

The sentence has been rewritten.

Round 2

Reviewer 2 Report

Dear Authors,

The ms is well improved overall. In particular, I'm pleased you were able to include the NanoDrop. I think this is an important addition and, as a result, this paper will be of wider interest in the scientific community. Although, without knowing the actual protein concentration of the two venoms (which is not possible, I know), showing a difference b/w the two venoms within NanoDrop is unhelpful for researchers who only use a NanoDrop.

A few minor issues (and suggested changes):

L204: mixturesis

L97: dupliates is missing a c. Also remove s.

L53-54: Really awkward and non-scientific. Change sentence to: "As such, venom components are of wide interest to many research groups and have been the subject of extensive research."

L50: 'less interesting'

L40: reword to 'each technique yields significantly different results.'

L46: Unless you can find a ref for pre-digestion being a function of venom, I'd suggest removing that. Also 'prey/predators' instead of 'victims'. See snake venom papers for how snake venom has been described previously. Also, to 'kill' is not necessarily a venom function, as no evolutionary pressure difference exists b/w a venom which immobilises a prey long enough to digest (alive) vs. a dead prey item. 

L67: space missing

L68: space missing

L73: '...we evaluated how...'

L150: 'crucial for studies involving venom but...'

L166: reword

L167: be more detailed to avoid confusion for reader. e.g. all analysed what? And instead of 'result' perhaps 'venom concentration' to be clear.

L170: These data

L186: extinction coefficient needs defining.

L189: not sure if 'inaccurate' is the right word here. Maybe 'variable'? Not sure on that, as I'm unfamiliar with that parameter. 

L207: unclear what is meant, as grammar wonky here: "from what these differences result"

Throughout: it's 'duplicate' and 'triplicate', not plural. e.g. correct way: Samples were run in duplicate/triplicate. 

Note: perhaps mention future research should test a known protein concentration across multiple methods.

L211: space b/w analyzedvenom

L230: 'of determining venom protein concentration'

L231: space b/w methodof

L228: perhaps add a sentence regarding the protein concentration differences read by a given method may not affect venom studies researching venoms with similar venom compositions and key amino acids (e.g. geographical variation, individual variation, sexual variation) but would matter more with inter-genera comparisons that are more likely to have larger differences in venom composition. 

Author Response

Dear Reviewer,

Thank you for the time you've devoted to our article. We are very grateful for your help in improving it. We have changed or rewritten all the fragments you indicated. We have also added fragments where required. We have detailed our response below:

L204: mixturesis

The error has been corrected.

L97: dupliates is missing a c. Also remove s.

The error has been corrected.

L53-54: Really awkward and non-scientific. Change sentence to: "As such, venom components are of wide interest to many research groups and have been the subject of extensive research."

The sentence has been changed.

L50: 'less interesting'

The error has been corrected.

L40: reword to 'each technique yields significantly different results.'

The sentence has been changed.

L46: Unless you can find a ref for pre-digestion being a function of venom, I'd suggest removing that. Also 'prey/predators' instead of 'victims'. See snake venom papers for how snake venom has been described previously. Also, to 'kill' is not necessarily a venom function, as no evolutionary pressure difference exists b/w a venom which immobilises a prey long enough to digest (alive) vs. a dead prey item. 

The sentence has been changed.

L67: space missing

The error has been corrected.

L68: space missing

The error has been corrected.

L73: '...we evaluated how...'

The error has been corrected.

L150: 'crucial for studies involving venom but...'

The sentence has been changed.

L166: reword

We rephrase whole paragraph.

L167: be more detailed to avoid confusion for reader. e.g. all analysed what? And instead of 'result' perhaps 'venom concentration' to be clear.

This sentence as a whole paragraph was rewritten.

L170: These data

The error has been corrected.

L186: extinction coefficient needs defining.

The sentence has been added.

L189: not sure if 'inaccurate' is the right word here. Maybe 'variable'? Not sure on that, as I'm unfamiliar with that parameter. 

The word "inaccurate" could be misleading, but "variable" also not seems to be the optimal choice. That is why we rephrase the sentence to:

 "However, in complex, unknown mixtures, it is hard to correctly estimate this parameter, and thus the calculated values can be very different from the true ones"

L207: unclear what is meant, as grammar wonky here: "from what these differences result"

The sentence has been changed to: "It is not entirely clear why these differences arise"

Throughout: it's 'duplicate' and 'triplicate', not plural. e.g. correct way: Samples were run in duplicate/triplicate. 

The error has been corrected.

Note: perhaps mention future research should test a known protein concentration across multiple methods.

The sentence has been added.

L211: space b/w analyzedvenom

The error has been corrected.

L230: 'of determining venom protein concentration'

The error has been corrected.

L231: space b/w methodof

The error has been corrected.

L228: perhaps add a sentence regarding the protein concentration differences read by a given method may not affect venom studies researching venoms with similar venom compositions and key amino acids (e.g. geographical variation, individual variation, sexual variation) but would matter more with inter-genera comparisons that are more likely to have larger differences in venom composition. 

The sentence has been added.